# Enhancing catalytic performance of dilute metal alloy nanomaterials

Mathilde Luneau[1], Erjia Guan[2], Wei Chen[3], Alexandre C. Foucher [4], Nicholas Marcella [2], Tanya Shirman[5,6], David M.A. Verbart[1], Joanna Aizenberg [1,5,6], Michael Aizenberg [6], Eric A. Stach [4], Robert J. Madix[5], Anatoly I. Frenkel [2,7] & Cynthia M. Friend [1,5✉]

Dilute alloys are promising materials for sustainable chemical production; however, their composition and structure affect their performance. Herein, a comprehensive study of the effects of pretreatment conditions on the materials properties of $Pd_{0.04}Au_{0.96}$ nanoparticles partially embedded in porous silica is related to the activity for catalytic hydrogenation of 1-hexyne to 1-hexene. A combination of in situ characterization and theoretical calculations provide evidence that changes in palladium surface content are induced by treatment in oxygen, hydrogen and carbon monoxide at various temperatures. In turn, there are changes in hydrogenation activity because surface palladium is necessary for $H_2$ dissociation. These $Pd_{0.04}Au_{0.96}$ nanoparticles in the porous silica remain structurally intact under many cycles of activation and deactivation and are remarkably resistant to sintering, demonstrating that dilute alloy catalysts are highly dynamic systems that can be tuned and maintained in a active state.

[1] Department of Chemistry and Chemical Biology, Harvard University, Cambridge, MA 02138, USA. [2] Department of Materials Science and Chemical Engineering, Stony Brook University, Stony Brook, New York 11794, USA. [3] Department of Physics and School of Engineering and Applied Sciences, Harvard University, Cambridge, MA 02138, USA. [4] Department of Materials Science and Engineering, University of Pennsylvania, Philadelphia, PA 19104, USA. [5] John A. Paulson School of Engineering and Applied Sciences, Harvard University, Cambridge, MA 02138, USA. [6] Wyss Institute for Biologically Inspired Engineering, Harvard University, Cambridge, MA 02138, USA. [7] Division of Chemistry, Brookhaven National Laboratory, Upton, New York 11973, USA. ✉email: friend@fas.harvard.edu

Heterogeneous catalysis plays an essential role in determining energy efficiency and greenhouse gas emissions in the chemical industry. Catalysis and catalytic processes account for 20–30% of the worldwide gross domestic product and 50 of the highest volume chemical processes account for more than 20 billion tons of $CO_2$ emission per year[1]. Clearly, there is a tremendous opportunity to reduce energy demand and $CO_2$ emission through improvements in catalytic processes. A major goal is to improve the selectivity towards desired products to avoid costly separation of by-products while also maintaining high activity. To achieve this goal, new catalytic materials with stable performance are required.

To this end, dilute bimetallic nanoparticles are being investigated as catalysts with the capability for high rates and selectivity[2–8]. The underlying principle is that a small amount of a reactive metal initiates the catalytic cycle, while the less reactive majority host material imparts higher selectivity to the overall process. For this principle to hold, the minority metal must be present and highly dispersed on the catalyst surface.

Dilute Pd/Au catalysts (Pd/Au ratio < 1) have been studied in particular for both selective oxidation[9–12] and selective hydrogenation reactions[11,13–17]. In these cases, Pd serves to activate either $O_2$ or $H_2$ to initiate the catalytic cycle since these processes do not readily occur on Au[18,19]. Hence, the presence of Pd on the surface of the catalyst is critical.

While improving selectivity is important, stable catalyst performance is also essential. Ideally, catalysts would remain stable over extended periods of time and deactivation would be reversible. Therefore, an essential part of a catalytic process is the activation processes used to generate and regenerate the active material. There are several methods used to pretreat catalyst materials prior to steady-state operation. Examples used for Pd/Au alloys are treatment in $H_2$[13,14,16] exposure to $O_2$[12], and flowing a mixture of $H_2$ and $O_2$ over the catalysts[11]. Generally, these treatments are empirically chosen without guiding principles. To truly realize the goal of designing efficient catalytic processes requires not only new classes of materials, but also understanding how catalyst activation can be controlled so as to create a robust system.

In this study, a set of guiding principles is derived from an extensive study of the effects of temperature and gas composition during pretreatment on the performance of $Pd_{0.04}Au_{0.96}$ nanoparticles embedded in so-called "raspberry colloid-templated" (RCT)-SiO$_2$. These specific catalyst materials were selected because they are active for selective hydrogenation of 1-hexyne to 1-hexene at high conversion[15].

Herein, this remarkable catalyst performance is related to the redistribution of Pd. A comprehensive set of studies using catalytic flow experiments, in situ microscopy, in situ spectroscopy and theoretical calculations are used to establish that Pd is redistributed by pretreatments in detail. Remarkably, $Pd_{0.04}Au_{0.96}$ RCT-SiO$_2$ is highly resistant to deactivation and sintering contrary to typical gold catalysts[20–23].

The challenge presented by this catalytic material is that Pd is atomically dispersed in the bulk in the most thermodynamically stable state of pristine dilute Pd–Au nanoparticles and is, therefore, unavailable for catalysis. Oxidative treatment provides stabilization of Pd at the surface as an oxide; judicious choice of a moderate reaction temperature in the hydrogenation reaction leads to high activity and sustained activity by kinetically trapping Pd on the catalyst surface in steady state; reduction in $H_2$ at high temperature (673 K) actually deactivates the catalyst as Pd redistributes itself into the bulk, since Pd–H binding is not sufficiently strong to stabilize surface Pd, and the steady-state coverage of H is low at this high temperature. The activity can be recovered by providing a thermodynamic driving force for Pd

segregation back to the surface. Treatment in CO at low temperature leads to partial recovery of activity, whereas treatment in $O_2$ at high temperature returns the catalyst to full activity. The redistribution of Pd in Au is reversible over many cycles of activation and deactivation; thus, the catalytic activity of the catalyst is completely tunable by using the adequate treatment. The specific catalysts investigated herein—$Pd_{0.04}Au_{0.96}$ nanoparticles RCT-SiO$_2$—are also resistant to sintering at high temperature.

## Results and discussion

**$Pd_{0.04}Au_{0.96}$ nanoparticles embedded in porous SiO$_2$.** Catalysts comprised of $Pd_{0.04}Au_{0.96}$ (4 atomic % of Pd) nanoparticles partially embedded in a highly structured matrix of SiO$_2$ were prepared using a published method[9,15,24]. The matrix contains 380 nm voids interconnected by 80 nm windows (Fig. 1a–c). The average diameter of the PdAu nanoparticles is $5.9 \pm 1.9$ nm after calcination in air at 773 K for 2 h, with Pd and Au homogeneously distributed in all nanoparticles analyzed (Supplementary Table 1, Fig. 1d–f and Supplementary Fig. 1)[15]. These materials are referred to as $Pd_{0.04}Au_{0.96}$ RCT-SiO$_2$, to signify the raspberry colloid templating procedure used to make the catalysts. We focus on the $Pd_{0.04}Au_{0.96}$ catalysts because previous work established that this composition yielded a highly selective and active catalyst after calcination at 773 K for hydrogenation of 1-hexyne to 1-hexene, which is used as a probe reaction in this work (Fig. 1g)[15].

**Pd is stable in the bulk in the absence of adsorbates.** Pd resides in the bulk of dilute reduced PdAu nanoparticles, based on measurements of its catalytic activity, supported by density functional theory (DFT) calculations. Since the size of the nanoparticles used in our experiments is too large for explicit DFT calculations, a Pd/Au(211) surface was used to probe the distribution of Pd, analogous to prior literature[25]. Such calculations indicate that Pd atoms are energetically favored to reside in the bulk of the material by 0.27 eV (Fig. 2a, b). The tendency of Pd atoms to reside in the subsurface or in the bulk of the Au host is due to the higher surface energy of Pd compared to that of Au[26,27]. Furthermore, the energy is lowest when Pd is dispersed as isolated atoms in the bulk; dimerization of Pd atoms is slightly unstable. For example, a Pd dimer in the 4th site is higher in energy than two separated Pd atoms by 0.05 eV (Supplementary Fig. 2). Moreover, the second Pd atom of this potential pair still strongly prefers to be in the bulk, indicating that Pd atoms in Au are atomically dispersed at low Pd concentrations below the surface except by random association.

Studies of the activity of $Pd_{0.04}Au_{0.96}$ RCT-SiO$_2$ catalysts confirm that Pd is preferably into the bulk and thus not available for reaction after pretreatment in He at 673 K based on the complete lack of activity for 1-hexyne hydrogenation at 363 K (Supplementary Fig. 3). Prior studies of Pd on Au(111) showed that Pd diffuses into the bulk at high temperature[18], consistent with our experiments and calculations. Thus, the surface of a pristine catalyst consists primarily of non-active Au atoms, necessitating methods for increasing and controlling the Pd surface concentration to activate the catalyst.

**Oxidation at high temperature stabilizes Pd at the surface.** To this goal, we found that prior oxidation of a partially deactivated $Pd_{0.04}Au_{0.96}$ RCT-SiO$_2$ catalyst via exposure to $O_2$ increases activity for selective hydrogenation of 1-hexyne (Fig. 2d). The activity of the catalyst in 1-hexyne hydrogenation progressively increases from conversions of 20, 23 and 30% as the temperature

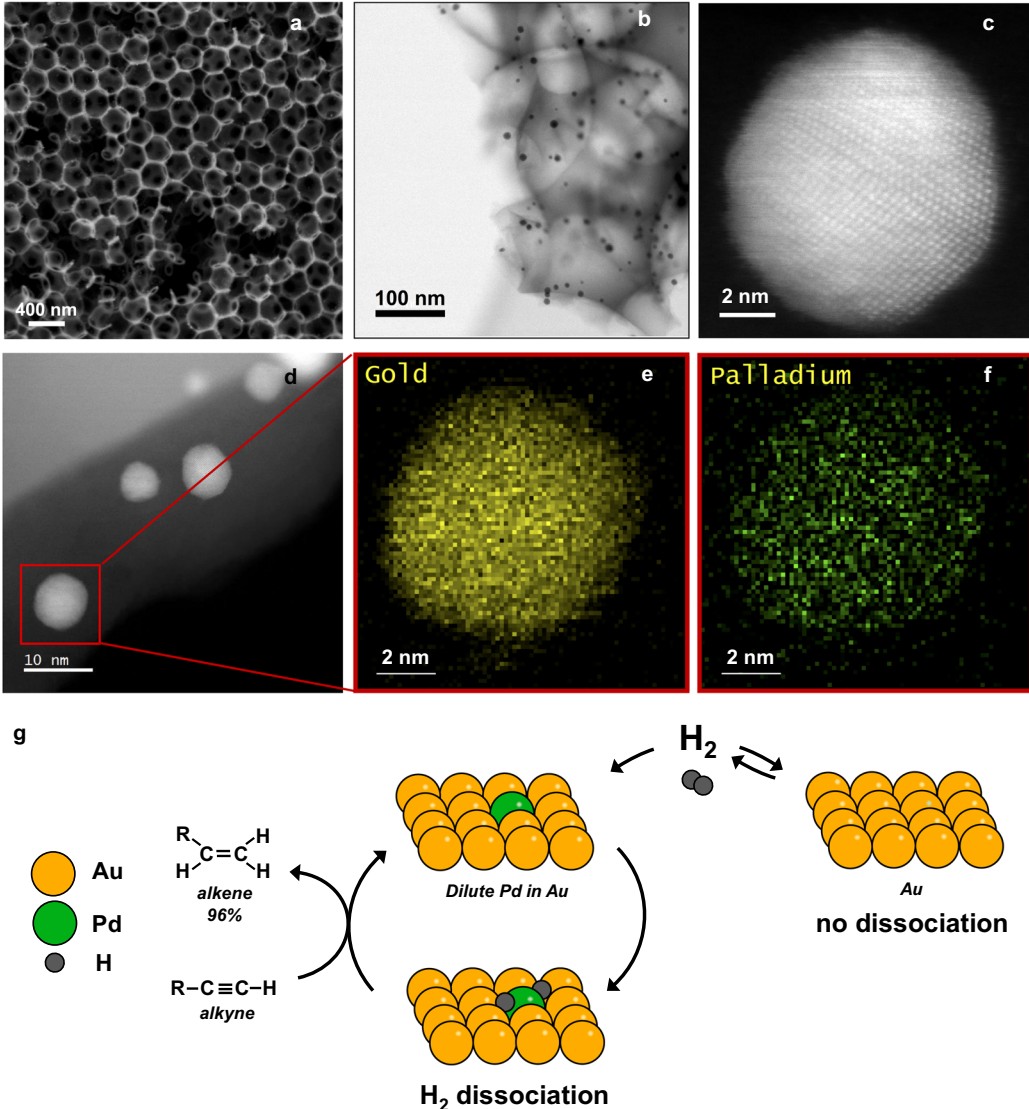

**Fig. 1 The Pd$_{0.04}$Au$_{0.96}$ nanoparticles are embedded in porous SiO$_2$. a** Scanning Electron Microscopy. **b, d** Transmission Electron Microscopy, **c** atomically resolved TEM images, and **e, f** energy-dispersive X-ray spectroscopy maps showing the distribution of Pd and Au in Pd$_{0.04}$Au$_{0.96}$ RCT-SiO$_2$ after calcination in air for 2 h at 773 K. **g** Schematic showing that Pd on the surface is critical for H$_2$ dissociation and alkyne hydrogenation whereas no reaction occurs on Au alone.

of the preceding exposure of the catalyst to O$_2$ increased from 323 K (10 h), 523 K (1 h) and 673 K (1 h), while the selectivity remained high, ~96% (Fig. 2d, Supplementary Table 2). Monometallic Au RCT-SiO$_2$ was tested in previous work after treatment at 773 K in O$_2$ and showed no activity under these reaction conditions[15].

The increased activity of the Pd$_{0.04}$Au$_{0.96}$ RCT-SiO$_2$ following pretreatment in O$_2$ is attributed to the stabilization of Pd on the surface driven by Pd–O bond formation. Prior studies of Au$_{30}$Pd$_{70}$(110) showed that Pd segregates to the surface upon exposure to O$_2$, forming a strained PdO phase[28]. The DFT calculations likewise show that formation of a surface oxide of Pd provides a thermodynamic driving force for the segregation of Pd on the surface. Notably, a single adsorbed O atom does not provide sufficient energetic stabilization to favor Pd on the surface; Pd bound to a single O atom on the surface is still unfavorable relative to Pd in the bulk separated from O on the surface Au by 0.22 eV (Fig. 2c). Formation of an extended Pd oxide phase does, however, favor Pd accumulation at the surface (Fig. 2c).

The variation of catalyst activity with the temperature used for O$_2$ treatment is largely attributed to kinetic effects. In order for PdO to form on the surface, Pd atoms must occasionally emerge to react with the oxygen. Over the range of temperatures investigated (298–773 K) the frequency of Pd atoms visiting the surface by random walk increases with the temperature. The overall thermodynamic stability of the oxide can also be affected by temperature. For example, at temperatures that are much higher than those we investigated, PdO will favor decomposition to release gas O$_2$.

Extended X-ray absorption fine structure (EXAFS) spectra of the Pd$_{0.04}$Au$_{0.96}$ RCT-SiO$_2$ catalyst measured at the Pd K edge provide additional evidence for the formation of Pd–O species on the surface after O$_2$ treatment at 673 K (Table 1, Supplementary Fig. 4, Supplementary Table 3). There is a clear Pd–O contribution to the spectrum with a coordination number of 0.2. The Pd–O distance (2.2 Å) is longer than in bulk PdO (2.01 Å)[29]. Long Pd–O bonds are attributed to chemisorbed surface oxygen, not oxygen in the PdO phase, consistent with XANES data (Supplementary Fig. 5). The Pd concentration at the

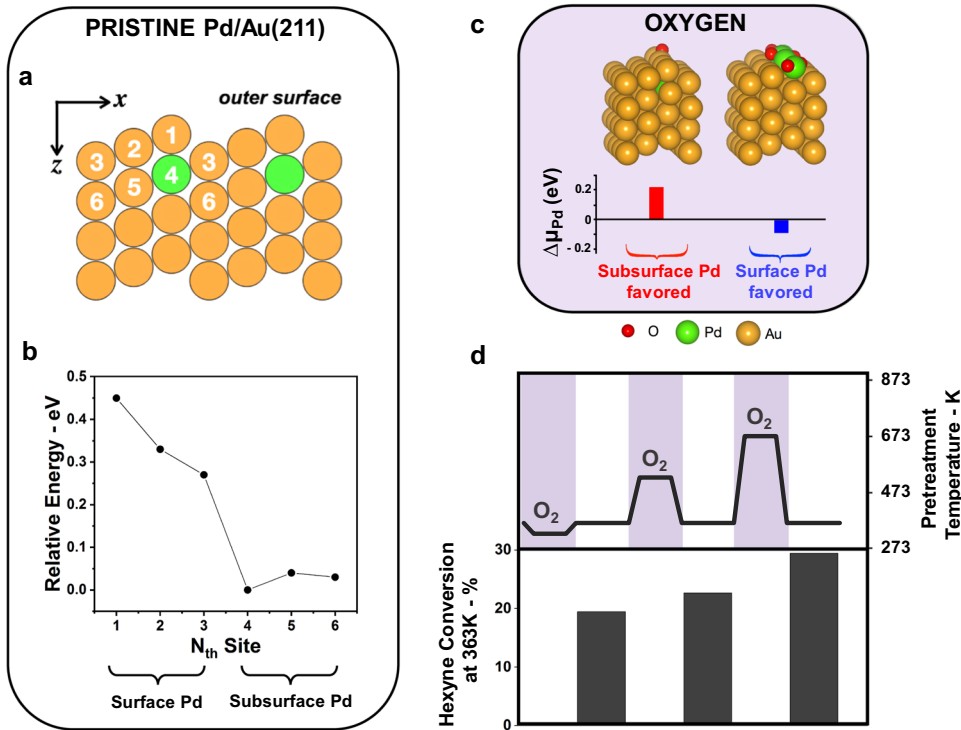

**Fig. 2 Oxidation at high temperature stabilizes Pd at the surface. a** A side view of the Au(211) surface with its site indexes of the top 6 sites labeled ($N$ = 1–6) and one Au atom at the 4th site substituted by Pd (green). **b** Pd is favored to be in the bulk in pristine Pd/Au(211) based on DFT calculations of the relative energies of structures with one Pd positioned at different sites. The energy zero is set at the lowest energy configuration ($N$ = 4). **c** Calculated chemical potential difference between Pd on the surface and Pd in the subsurface ($\Delta\mu_{Pd} = \mu_{Pd@surf} - \mu_{Pd@sub}$) of the DFT-calculated most stable structures of Pd/Au(211). Each bar corresponds to the overlayer structure above it. Subsurface Pd is favored in the presence of isolated O while Pd is stabilized as an oxide at the surface with increased O concentration. **d** Exposure of the catalyst to $O_2$ (20% $O_2$ in He at atmospheric pressure i.e., $pO_2$ = 0.2 bar) at progressively higher temperature—323 K (10 h), 523 K (1 h) and 673 K (1 h)—yields catalysts with progressively higher activity. Reaction conditions: 1% 1-hexyne; 20% $H_2$ in He balance; $T$ = 363 K; $m_{cat}$ = 20 mg; Total flow rate = 50 mL min$^{-1}$; GHSV = 3800 h$^{-1}$.

**Table 1 EXAFS structure parameters representing RCT silica supported Pd$_{0.04}$Au$_{0.96}$ nanoparticles after pretreatments of $O_2$, $H_2$ and CO.**

| Pretreatment | Pd-O | | Pd-Au | | Pd-Pd | |
|---|---|---|---|---|---|---|
| | CN | R (Å) | CN | R (Å) | CN | R (Å) |
| $O_2$ (673 K) | 0.2(1) | 2.20(2) | 11.5(4) | 2.825(2) | 0.2(1) | 2.80(2) |
| $H_2$ (673 K) | – | | 11.6(5) | 2.825(2) | 0.2(1) | 2.82(2) |
| C0 (298 K) | – | | 11.3(5) | 2.825(2) | 0.2(1) | 2.78(2) |
| $O_2$ (673 K, recover) | 0.2(1) | 2.20(2) | 11.5(5) | 2.825(2) | 0.2(1) | 2.80(2) |

Fitting ranges: k-range: 2.0–13.6 Å$^{-1}$, R-range: 1.35–4.00 Å. Uncertainties in the last significant digits are given in parentheses. Detailed EXAFS analysis including Debye-Waller factors and energy correction terms are in the SI.
N, coordination number; R, distance between X-ray absorbing and backscattering atoms.

surface is on average ~4% based on quantitative analysis of EXAFS data. Specifically, the Pd–O coordination number of 0.2 indicates that up to 20% of the total Pd is drawn to the surface after oxidation assuming a stoichiometry of PdO (calculations in SI, Supplementary Eq. 1). Such low Pd surface concentration prevents the alloy from presenting a Au–Pd core-shell structure sometimes reported in the literature[30,31]. EXAFS was necessary to obtain an estimate of the Pd surface concentration because other methods, e.g., X-ray photoelectron spectroscopy, is not sufficiently sensitive due to charging of the sample, the extremely low amount of Pd (~0.06 wt.%) and the fact that the Pd3d and Au4d$_{5/2}$ regions overlap.

Pd is clearly present on the surface after $O_2$ treatment and subsequent reduction based on the fact that H–D exchange occurs in the range of 363–673 K for a mixture of $H_2$ and $D_2$ (Supplementary Fig. 6). Notably, no H–D exchange occurs for the pure Au catalyst. Because the reaction is not at equilibrium a more detailed interpretation of the temperature dependence is not possible.

**Adsorbed hydrogen does not stabilize Pd at the surface.** In contrast to the effects produced by treatment of the catalyst in $O_2$, pre-treatment of the Pd$_{0.04}$Au$_{0.96}$ RCT-SiO$_2$ catalyst in hydrogen ($p(H_2)$ = 0.2 bar) at high temperature is detrimental to catalyst activity. Only moderate deactivation occurs for treatment at 523 K (1 h) in $H_2$, but a more substantial loss occurs at 673 K (30 min) (Fig. 3). Remarkably, the catalyst can be subsequently reactivated by treatment in $O_2$ at 773 K (Fig. 3). The selectivity towards 1-hexene remained high for all conditions (Supplementary Table 2). The dependence on temperature used for $H_2$ treatment is again attributed to kinetic effects; in this case, Pd more rapidly dissolves into the bulk of the nanoparticle during pretreatment at higher temperature, causing significant loss of Pd from the surface and, therefore, diminished activity.

DFT calculations and EXAFS corroborate the assertion that subsurface Pd is preferred in a hydrogen gas environment. In the presence of a single adsorbed H, the Pd is still thermodynamically favored to reside in the subsurface (Fig. 3b). The energy required to reposition a Pd atom from the subsurface to the top row adjacent to a H atom is endothermic by 0.27 eV. Upon formation of a surface hydride at increased H coverages, the Pd atoms still prefer subsurface sites, though the energy required to relocate to

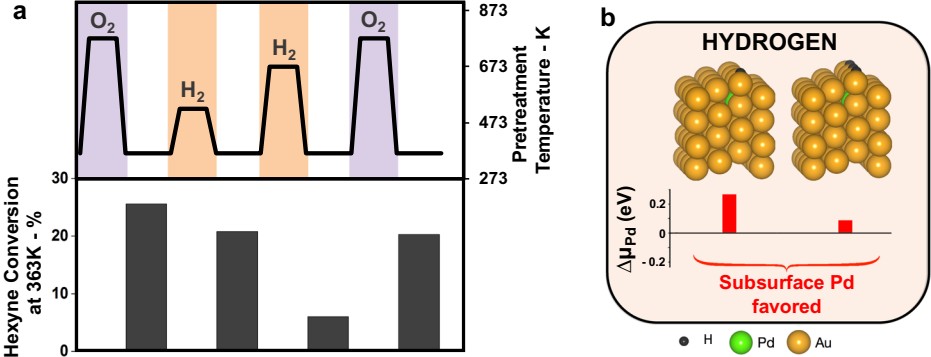

**Fig. 3 Adsorbed hydrogen does not stabilize Pd at the surface. a** Oxygen treatment at 773 K (30 min) activates the catalyst as shown by the black histograms. Hydrogen treatment of this catalyst at 523 K (1 h) and 673 K (30 min) leads to a decrease in conversion when the catalyst is re-exposed to reaction conditions at 363 K. Conversion can be recovered by high-temperature treatment in oxygen. **b** Pd prefers the subsurface in the presence of isolated H and with increased H concentration, as represented by the DFT-calculated most stable structures of Pd/Au(211) surface upon adsorption of isolated H and with increased H concentration. Hexyne reaction: 1% 1-hexyne; 20% $H_2$ in He balance; $T = 363$ K; $m_{cat} = 20$ mg; Total flow rate = 50 mL min$^{-1}$; GHSV = 3800 h$^{-1}$.

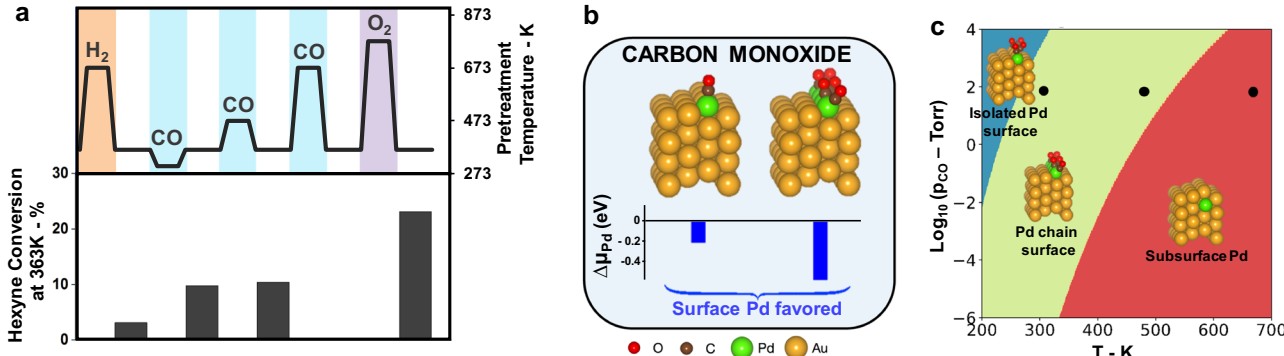

**Fig. 4 The effect of CO on Pd distribution depends on temperature. a** $Pd_{0.04}Au_{0.96}$ RCT-SiO$_2$ catalyst pretreatment with specific gases at different temperatures and the 1-hexyne conversion resulting therefrom. Initial exposure to H$_2$ reduces conversion. Conversion is partially recovered after CO pretreatment at low temperatures (1 h). At high temperature, all conversion is lost. High-temperature O$_2$ treatment leads to a full recovery of the initial conversion. Reaction: 1% 1-hexyne; 20% H$_2$ in He balance; $T = 363$ K; $m_{cat} = 20$ mg; Total flow rate = 50 mL min$^{-1}$; GHSV = 3800 h$^{-1}$. **b** Pd is stabilized at the surface in the presence of isolated CO and high coverages of CO. DFT-calculated most stable structures of PdAu(211) surface upon adsorption of CO molecules. **c** At low temperature and high pressures of CO, Pd is stabilized as a single atom at the surface. At higher temperatures, Pd is stabilized as a chain at the step of the surface. At low pressures and high temperatures, CO is not able to stabilize Pd at the surface and Pd resides in the subsurface. The black dots show the three different conditions relevant to the experimental conditions. At room temperature and 473 K, Pd is at the surface, at 673 K, Pd is present at the subsurface.

the surface decreases from 0.27 to 0.09 eV. The lack of detectable Pd–O signal in the EXAFS spectra is also consistent with Pd dissolution (Table 1). Analysis of the Pd K-edge XANES region shows no detectable changes in Pd oxidation state (Supplementary Fig. 5).

The observed decrease in activity of the $Pd_{0.04}Au_{0.96}$ RCT-SiO$_2$ after pretreatment in H$_2$ at 673 K is consistent with prior studies with more Pd-rich catalysts. For example, the activity for acetylene hydrogenation by $Pd_xAu_{1-x}$ catalysts supported on SiO$_2$ ($0.46 > x > 0.15$) was decreased by prior reduction in hydrogen[32]. $Pd_{0.15}Au_{0.85}$ was still active after treatment in hydrogen at 423 K, but was almost completely deactivated after treatment at 573 K. Thus, the effect of H$_2$ on the Pd distribution in these alloy catalysts appears general.

**The effect of CO on Pd distribution depends on temperature.** Treatment in flowing CO also strongly effects the catalyst performance. A $Pd_{0.04}Au_{0.96}$ RCT-SiO$_2$ catalyst deactivated by H$_2$ can also be partially reactivated by treatment in CO, even at room

temperature (pCO = 0.1 bar) (Fig. 4a). The 1-hexyne conversion at 363 K increased from 3% after H$_2$-induced deactivation to 10% when exposed to CO at room temperature (1 h) (Fig. 4a). The selectivity towards 1-hexene was again not affected by the pretreatment and remained high (Supplementary Table 2).

This partial reactivation is again attributed to adsorbate-induced stabilization of surface Pd. The deduction that CO stabilizes Pd on the surface is supported by DFT studies (Fig. 4b), as the structure with CO bound to a single Pd atom on the (211) surface is 0.21 eV more stable than the structure with Pd in the subsurface region and CO bound to Au alone at the surface. Furthermore, short chains of Pd atoms at the step edge form a very stable structure at higher CO coverages (Fig. 4b). Moreover, EXAFS spectra showed that the Pd–Pd bonding distance decreased to 2.78 Å after pretreatment in CO at 298 K. The shorter (by 0.04 Å, compared to the pretreatment in H$_2$, Table 1) Pd–Pd distance is attributed to migration of more Pd atoms to the surface induced by CO. An analogous contraction of the Au–Au bond length was previously reported for surface atoms on a Au nanoparticle[33]. No detectable changes in oxidation state

could be observed in the Pd–K edge XANES region after the CO treatment (Supplementary Fig. 5).

Increasing the temperature of the CO treatment continued to improve the catalytic activity up to 473 K (1 h), but above that temperature activity decreased and was lost after CO treatment at 673 K (1 h). This loss of activity is explained by carbon deposition from CO decomposition (Supplementary Fig. 7) as well as the lower steady-state coverage of CO under these conditions. Investigation of the thermodynamic stability of Pd bound to CO as a function of temperature and pressure using DFT predicts that Pd will be segregated on the surface at 300 and 473 K under 0.1 bar (75 Torr) of CO, but that Pd will remain in the subsurface at 673 K (Fig. 4c). The temperature effect observed for CO is thus attributed to changes in the steady-state CO coverage with temperature combined with the thermodynamic considerations. The stabilization of Pd at the surface by CO is consistent with previous studies of AuPd(100)[10] and AuPd(110)[34] which show that Pd is stabilized at the surface even at very low CO pressures. For example, exposure of CO ($<10^{-3}$ Torr) to Au-terminated AuPd(100) induces Pd segregation to the surface. At higher pressure ($>0.1$ Torr), Pd pairs were observed. Similarly, Pd segregated to the surface on AuPd(110) even at CO pressures as low at $10^{-6}$ Torr[34]. Likewise, exposure to CO at room temperature induces Pd segregation in $Pd_{0.05}Au_{0.95}/Al_2O_3$ catalysts after reduction at 773 K[35].

**The catalyst does not measurably sinter**. Summarizing the effects of the gaseous environment on the activity of the alloy catalysts, the differences in catalytic performance induced by the treatments in $O_2$, $H_2$ and CO are attributed solely to a change in surface concentration of Pd. No change in the overall catalyst structure is observed in the TEM images collected in situ during exposure of the catalyst to $O_2$ at 773 K or to $H_2$ at 673 K (Fig. 5a–c). Further, the $Pd_{0.04}Au_{0.96}$ particle sizes and shapes do

not measurably change, and the silica remains highly structured under these conditions. These observations indicate that the differences in catalytic performance after the two different pre-treatments are not due to sintering of the nanoparticles nor loss of structure in silica.

The Pd distribution measured after the pretreatments persist under reaction conditions; however the Pd is reduced—no oxidized Pd is detected—based on in situ EXAFS analysis (Supplementary Fig. 8). The Pd predominantly remains on or near the surface under catalytic operation for the case of $O_2$ pretreatment at high temperature based on the measured R (Pd–Pd) of 2.70 Å ± 0.03 (Supplementary Table 4). In contrast under reaction conditions after $H_2$ treatment, the R(Pd–Pd) is (2.77 Å ± 0.02), consistent with Pd re-dissolved in the bulk.

**The catalytically active state is repeatedly regenerated**. The remarkable robustness of the $Pd_{0.04}Au_{0.96}$ RCT-SiO$_2$ catalyst is, indeed, demonstrated by the resistance to sintering and the ability to repeatedly restore the activity of the catalyst by gas phase treatments. In previous work, the catalyst proved to be stable for more than 30 h of operation at 363 K after initial carbon deposition[15]. During the scope of this study, the catalyst underwent more than 10 different treatments at high temperature in either CO, $H_2$ or $O_2$ environments and was exposed to reaction conditions for a total time of ~170 h on stream. Post-reaction TEM measurements indicate that the silica structure remained unchanged (Fig. 5d, e) and only a small increase in the distribution of particle size was observed (Fig. 5f). The average size is, however, not significantly higher than on the as prepared catalyst before reaction (5.9 ± 1.9 nm vs. 6.5 ± 2.1 nm). The catalytic activity of the catalyst was fully recoverable by oxygen treatment at high temperature, both after partial loss of activity due to hydrogen treatment at 663 K (Fig. 3a) and total loss of activity due to CO treatment at 663 K (Fig. 4a). EXAFS spectra showed nearly identical structure parameters with

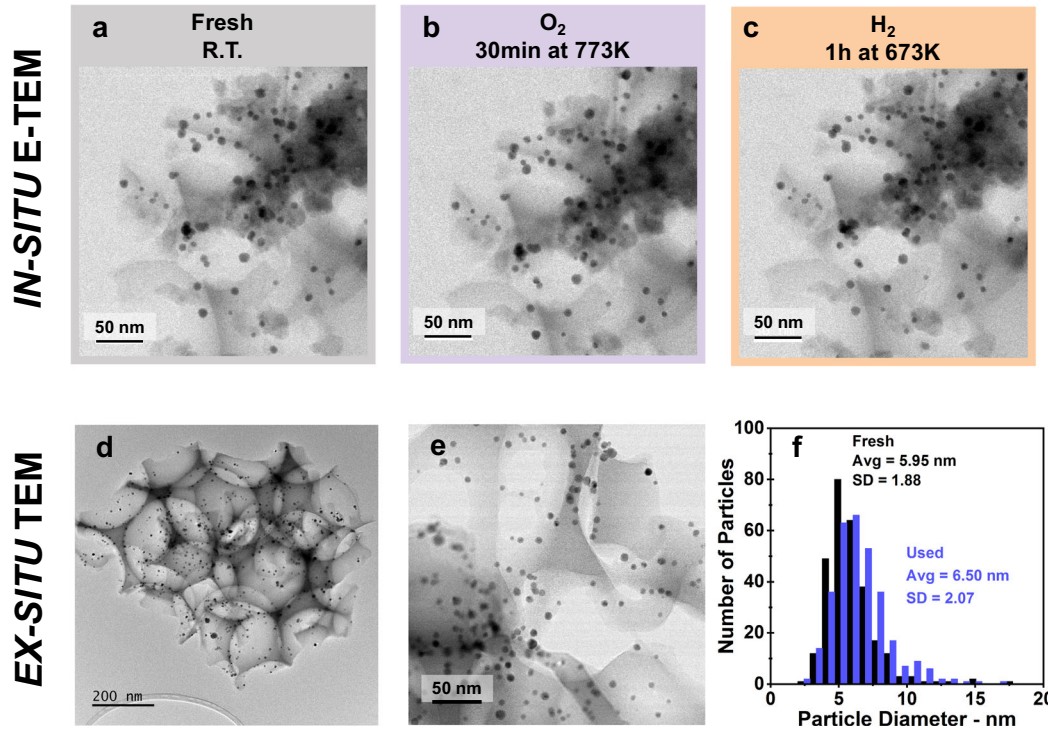

**Fig. 5 The catalyst does not sinter at high temperature.** The silica structure and the shape of **a** as-prepared $Pd_{0.04}Au_{0.96}$ RCT-SiO$_2$ are indistinguishable from those exposed to **b** $O_2$ at 773 K for 30 min and to **c** $H_2$ at 673 K for 1 h as shown by in situ TEM images. **d** The silica structure is intact and **e**, **f** the particles do not sinter after more than 10 cycles of high-temperature treatments and catalytic flow reaction at 363 K. Only a slight increase of the average particle size diameter is observed by TEM measurements of the nanoparticles in used $Pd_{0.04}Au_{0.96}$ RCT-SiO$_2$.

those of initially $O_2$ treated samples (Table 1, Supplementary Fig. 4, Supplementary Table 3).

In summary, this work illustrates how the understanding gained by a combination of DFT and in-depth experimental studies can be used to manipulate the surface composition of dilute alloy catalysts, and hence it catalytic activity, by controlling the gas-phase composition and of temperature. Catalytic flow experiments, in situ spectroscopy, in situ microscopy and theoretical calculations all corroborate the fact the Pd surface content is different in various gas environments at at various temperature. Key to achieving high activity is the creation a surface alloy with a significant number of sites to initiate the reaction; in the case of dilute Pd/Au catalysts for hydrogenation Pd must be present. Since Pd is favored to reside in the bulk on pristine Pd/Au, gas pretreatment must be used to provide a thermodynamic driving force for segregation of Pd on the surface. Either $O_2$ or CO in the gas impart sufficient thermodynamic stabilization for Pd to reside on the surface, whereas $H_2$ does not. The temperature for pretreatment is also important; higher temperatures lead to more rapid rearrangement of the alloy, but may also ultimately lead to low steady-state coverages of adsorbates.

Finally, the ability of the catalyst itself to remain structurally intact under many cycles of activation and deactivation is important and noteworthy. The partially embedded $Pd_{0.04}Au_{0.96}$ nanoparticles in the RCT-$SiO_2$ are remarkably resistant to sintering and, therefore, can be reactivated at elevated temperature repeatedly. These materials show that dilute alloy catalysts are highly dynamic systems that can be tuned and maintained in their most active states. The development of general design principles involving the activation methods for dilute alloy catalysts are thus critically important to their development.

## Methods

**$Pd_{0.04}Au_{0.96}$ RCT-$SiO_2$ preparation**. Styrene, acrylic acid, ammonium peroxodisulfate, gold (III) chloride hydrate (HAuCl$_4$, 99.995%), palladium (II) nitrate hydrate (Pd(NO$_3$)$_2$, 99.9%), sodium borohydride (NaBH$_4$, 99%), poly-vinylpyrrolidone (PVP, MW 10 K), sodium citrate, 2-aminoethanethiol hydro-chloride (AET, 98%), N-Ethyl-N'-(3-(dimethylamino)propyl) carbodiimide hydrochloride (EDAC, ≥99.0%) and 2-(N-morpholino)ethanesulfonic acid (MES, >99.5%), tetraethyl orthosilicate (TEOS), nitric acid (HNO$_3$ (aq), 67–70% w/w), hydrochloric acid (HCl (aq), 36.5−38.0% w/w and 0.1 M), hydrofluoric acid (HF, 50 %), acetone and sand (white quartz, 50–70 mesh particle size) were purchased from Sigma-Aldrich. Ethanol was obtained from Koptec. ICP-MS calibration samples of Au (10 PPM) and Pd (10 PPM) were obtained from Inorganic Ventures. All the chemicals were used as received. Triply distilled deionized (DI) water (18 MΩ) was used in all experiments. All glassware and teflon-coated magnetic stir bars used in the metal nanoparticle synthesis were thoroughly cleaned in aqua regia (3 parts HCl, 1 part HNO$_3$) (Caution: highly corrosive) and rinsed in DI water.

The synthesis of PVP-capped bimetallic $Pd_{0.04}Au_{0.96}$ nanoparticles (NPs) and raspberry colloids were reported in our previous publications[9,15]. In general, citrate capped Au nanoparticles (~5 nm) were prepared by reduction of HAuCl$_4$ with sodium borohydride NaBH$_4$ in DI water. As synthesized Au nanoparticle solution (40 mL) was used for the synthesis of $Pd_{0.04}Au_{0.96}$ by adding to it 5 mL of ascorbic acid aqueous solution (0.1 M) and 150 µL of Pd(NO$_3$)$_2$ aqueous solution (10 mM). The reaction mixture was stirred for 12 h at room temperature and stored at 4 °C. Carboxylic acid-functionalized polystyrene colloids (PS-COOH) with diameter of ~340 nm were synthesized by surfactant free emulsion polymerization, using acrylic acid as co-monomer and ammonium peroxodisulfate as an initiator following a recipe from the literature[36]. Raspberry colloids were synthesized by adding specific amounts of $Pd_{0.04}Au_{0.96}$ nanoparticles to the colloidal dispersion of thiol-modified polystyrene colloids (PS-SH)[9,15]. Typically for ~1% metal loading, 2.5 mL of the $Pd_{0.04}Au_{0.96}$ solution was added to 1 mL of 1 wt.% PS-SH colloidal dispersion in DI water. The dispersion was stirred for 2 h, washed three times with water using centrifugation (9500 rpm for 40 min), and re-dispersed in water to give ~5 wt% PS@$Pd_{0.04}Au_{0.96}$ raspberry colloids. The backfilling method used to form RCT SiO$_2$-based structures was described in detail in our previous publications[24,37]. In general, the raspberry colloidal dispersion was dried at 65 °C and then backfilled with prehydrolyzed TEOS solution. The backfilled samples were dried and finally calcined at 500 °C in air for 2 h to remove polymer colloids and organic volatiles, and to solidify the matrix into SiO$_2$.

**Catalytic tests**. Catalytic studies were carried out in a continuous-flow reactor at atmospheric pressure. $Pd_{0.04}Au_{0.96}$ RCT-$SiO_2$ was crushed and sieved to obtain

particle sizes diameter 100 < dp < 300 µm. The catalyst was then mixed with quartz sand and loaded into a tubular quartz reactor tube (internal diameter: 1 cm). Gas phase 1-hexyne hydrogenation was carried out at 363 K with a feed gas mixture of 1-hexyne (1%), H$_2$ (20%) in a balance of He with a total flow rate of 50 mL min$^{-1}$. Pretreatments in oxygen were carried out with 20% O$_2$ in He balance with at total flow rate of 50 mL min$^{-1}$(ramp 10 K min$^{-1}$). Pretreatments in hydrogen were carried out with 20% H$_2$ in He balance with a total flow rate of 50 mL min$^{-1}$ (ramp 10 K min$^{-1}$). Pretreatments in carbon monoxide were carried out with 10% CO in He balance with at total flow rate of 25 mL min$^{-1}$ (ramp 10 K min$^{-1}$). Catalytic performance was measured using a GC/MS (Agilent Column HP-Plot/Q). HD exchange experiments were performed with 20% H$_2$ and 1% D$_2$ in He balance with a total flow rate of 50 mL min$^{-1}$ and monitored with an online mass spectrometer (MS). Temperature-programmed oxidation was carried out with 20% O$_2$ in He at 773 K with a total flow rate of 20 mL min$^{-1}$ (ramp 10 K min$^{-1}$).

**DFT calculations**. DFT calculations were performed using VASP[38] with PAW potentials[39,40] and GGA-PBE[41] exchange-correlation functional. DFT-TS method[42] was used to include the van der Waals interactions. The kinetic energy cutoff of the plane-wave basis sets was 400 eV. The relaxed lattice constant of Au (4.11 Å) is close to the experimental value (4.08 Å). The Au(211) and Pd/Au(211) surfaces were modeled by slabs of 12 atomic sites separated by 15 Å of vacuum space. A Γ-centerd $8 \times 5 \times 1$ k-point mesh[43] was used for the supercells (7.12 and 11.63 Å in the x and y directions, respectively). For each molecule, several adsorption configurations were calculated to find the most stable structures. Atoms above the bottom four layers of Au were relaxed to a force threshold of 0.01 eV/Å. The chemical potential of Pd, in the case of m adsorbates on the surface, is calculated as $\mu_{Pd} = [E(ads_m/Au_{N-n}Pd_n) - E(ads_m/Au_N)]/n$, where N is the total number of metal atoms in the supercell, and n is the number of Pd dopant on the surface or in the subsurface. The difference between the lowest chemical potential of Pd on the surface ($\mu_{Pd@surf}$) and of Pd in the subsurface ($\mu_{Pd@sub}$), calculated as $\Delta\mu_{Pd} = \mu_{Pd@surf} - \mu_{Pd@sub}$, was used to identify the preference of Pd distribution at a given concentration of surface adsorbate. Due to a large number of possible configurations considered, the zero-point energy contribution was ignored. In the calculation of thermodynamic phase diagram, the standard entropy of CO was acquired from the *NIST* database[44].

**Microscopy**. Transmission electron microscopy was performed with two micro-scopes (JEOL JEM-F200 and an aberration-corrected JEOL NEOARM), both operating at 200 kV and in scanning transmission electron microscopy (STEM) mode. For ex situ studies, the sample was diluted in isopropanol and deposited on a lacey carbon film on copper grids. For in situ studies, an environmental holder for gas flow experiments manufactured by Hummingbird Scientific was used. The sample was enclosed in a microchip, made of two SiN windows and a micro electro-mechanical system for temperature control. The mass flow of gases was controlled using a gas system and a software provided by the same company.

Energy-dispersive X-ray spectroscopy (EDS) was performed with JEOL NEOARM and the maps were obtained with DigitalMicrograph by Gatan Inc (pixel time = 0.04 s and pixel size = 1.4 Å).

**X-ray absorption spectroscopy**. X-ray absorption spectroscopy (XAS) experiments at Pd K edge (24350 eV) were performed at the ISS beamline (8-ID), NSLS II, Brookhaven National Laboratory. The beam was monochromatized by a Si(111) high heat load double crystal monochromator. The samples in the form of fine powders were loaded into a capillary flow cell which allows in situ experiments.

*Pretreatments*: In the in situ experiments, the samples were exposed to flowing O$_2$ (20% balanced with He, 20 mL/min) at 298 K, followed by a temperature ramp to 673 K in 40 min and hold for 1 h. The cell was then cooled down to 298 K for the data collection. The procedures for H$_2$ pretreatment at 673 K were identical to those of O$_2$ pretreatment. The data representing the structure of the samples after CO pretreatment was recorded after 1-h treatment in CO at 298 K. The O$_2$ recovery experiments were performed after the samples went through O$_2$, H$_2$, CO pretreatments at 673 K and were then exposed to flowing O$_2$ at 673 K for 1 h and the data was collected at 298 K after the recovery.

*Reaction conditions*: In the in situ experiments, the sample was, in this order, pretreated with 20% O$_2$/He at 673 K, exposed to 1-hexyne at room temperature, exposed to 1-hexyne and H$_2$ at room temperature, and exposed to 1-hexyne and H$_2$ at 363 K. The sequence was repeated, except the O$_2$ in the first step was replaced with H$_2$.

The spectra were collected in fluorescence mode under in situ conditions and at least twenty spectra were merged after alignment for each analysis to ensure good signal-to-noise ratio. Analysis of the EXAFS data was carried out with the software ATHENA and ARTEMIS of the DEMETER package. Amplitude reduction factors $S_0^2$ (0.87 ± 0.02) were obtained from the EXAFS data fits for Pd foil spectrum and fixed to those values for the determination of coordination numbers in the $Pd_{0.04}Au_{0.96}$ RCT-$SiO_2$.

## Data availability

Authors can confirm that all relevant data are included in the paper and/or its supplementary information files

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

## Acknowledgements

This work was supported as part of the Integrated Mesoscale Architectures for Sustainable Catalysis (IMASC), an Energy Frontier Research Center funded by the U.S. Department of Energy, Office of Science, Basic Energy Sciences under Award #DE-SC0012573. ML gratefully acknowledges support from TOTAL. This research used 8-ID (ISS) beamline of the National Synchrotron Light Source, a U.S. DOE Office of Science User Facility operated for the DOE Office of Science by Brookhaven National Laboratory (BNL) under Contract No. DE-SC0012704. Work was also carried out in part at the Singh Center for Nano-technology at the University of Pennsylvania which is supported by the National Science Foundation (NSF) National Nanotechnology Coordinated Infrastructure Program grant NNCI-1542153. Additional support to the Nanoscale Characterization Facility at the Singh Center has been provided by the Laboratory for Research on the Structure of Matter (MRSEC) supported by the National Science Foundation (DMR-1720530).

## Author contributions

C.M.F., A.I.F., E.S., and R.J.M. guided the research and edited the manuscript, J.A. and M.A. participated in discussions and edited the manuscript, M.L., W.C., and E.G. wrote the manuscript, T.S. synthesized the catalyst, M.L. performed the catalytic tests. N.M., A.C.F., E.G., M.L., D.M.A.V., E.S., and A.I.F. carried out the XAS runs, N.M. and E.G. analyzed the data, W.C. carried out the theoretical calculations, A.C.F. collected the TEM images.

## Competing interests

The authors declare no competing interests.
