## [Peer Review File · Communications Chemistry]

This manuscript has been previously reviewed at another Nature Research journal. This document only contains reviewer comments and rebuttal letters for versions considered at Communications Chemistry.

REVIEWERS' COMMENTS:

Reviewer #1 (Remarks to the Author):

I am happy that my comments have been addressed and the paper is now suitable for publication

Reviewer #2 (Remarks to the Author):

I am happy to see that the authors provide in situ EXAFS data to support their main conclusions. I am satisfied with most of the responses to the reviewers' concerns. The manuscript now presents improved quality and is suitable for publication.

Reviewer #3 (Remarks to the Author):

This manuscript remains a very good characterization study. While I personally believe that the results are somewhat over-sold, researchers familiar with this reaction and bimetallic system will immediately recognize this and give the paper its due attention. The clear characterization of the chemistry associated with Pd mobility make this a valuable contribution to the literature. It is appropriate for Chem Commun.

There is one relatively small change that should be made prior to publication. The manuscript claims that Pd is "more stable" in the bulk than on the surface; this is largely based on a 0.05 eV difference in the DFT slab calculations. While the data show that, indeed, the sub-surface Pd is 0.05 eV lower in energy, this difference remains within reasonable DFT errors and is therefore probably not significant. I can live with modifying these statements being modified to "slightly more stable".

The manuscript would also benefit from clarifying that the "slightly" is an important property of the system. It is only because the energy differences are small that the Pd-O bond energy is large enough to pull the Pd to the surface. Similarly, the Pd remains on the surface to do the catalysis (at least initially) after O removal BECAUSE there is only a weak driving force for it to migrate back into the bulk. The exchange is due to these small energy differences. If Pd was much more stable in the bulk, it would stay there during oxidation or it would likely migrate back quickly after O removal. Readers would benefit from having clear statements to this effect.

REVIEWERS' COMMENTS:

Reviewer #1 (Remarks to the Author):

I am happy that my comments have been addressed and the paper is now suitable for publication

Response to reviewer #1: We thank the reviewer for the positive review. No changes to the manuscript were made.

Reviewer #2 (Remarks to the Author):

I am happy to see that the authors provide in situ EXAFS data to support their main conclusions. I am satisfied with most of the responses to the reviewers' concerns. The manuscript now presents improved quality and is suitable for publication.

Response to reviewer #2: We thank the reviewer for the positive review. No changes to the manuscript were made.

Reviewer #3 (Remarks to the Author):

This manuscript remains a very good characterization study. While I personally believe that the results are somewhat over-sold, researchers familiar with this reaction and bimetallic system will immediately recognize this and give the paper its due attention. The clear characterization of the chemistry associated with Pd mobility make this a valuable contribution to the literature. It is appropriate for Chem Commun.

There is one relatively small change that should be made prior to publication. The manuscript claims that Pd is "more stable" in the bulk than on the surface; this is largely based on a 0.05 eV difference in the DFT slab calculations. While the data show that, indeed, the sub-surface Pd is 0.05 eV lower in energy, this difference remains within reasonable DFT errors and is therefore probably not significant. I can live with modifying these statements being modified to "slightly more stable".

The manuscript would also benefit from clarifying that the "slightly" is an important property of the system. It is only because the energy differences are small that the Pd-O bond energy is large enough to pull the Pd to the surface. Similarly, the Pd remains on the surface to do the catalysis (at least initially) after O removal BECAUSE there is only a weak driving force for it to migrate back into the bulk. The exchange is due to these small energy differences. If Pd was much more stable in the bulk, it would stay there during oxidation or it would likely migrate back quickly after O removal. Readers would benefit from having clear statements to this effect.

Response to reviewer #3: The statement made about Pd being more stable in the subsurface in pristine PdAu(211) is based on a difference of 0.27 eV (not 0.05 eV) which is significant. The

value mentioned by the reviewer (0.05 eV) only refers to the dimerization of Pd – not the energy difference between surface and subsurface Pd.

We now have added the stabilization energy to be more explicit in the manuscript on page 3 last paragraph line 7

“Such calculations indicate that Pd atoms are energetically favored to reside in the bulk of the material **by 0.27 eV** (Figure 2a, b).”

--